# Bevacizumab or PARP-Inhibitors Maintenance Therapy for Platinum-Sensitive Recurrent Ovarian Cancer: A Network Meta-Analysis

**DOI:** 10.3390/ijms21113805

**Published:** 2020-05-27

**Authors:** Michele Bartoletti, Giacomo Pelizzari, Lorenzo Gerratana, Lucia Bortot, Davide Lombardi, Milena Nicoloso, Simona Scalone, Giorgio Giorda, Gustavo Baldassarre, Roberto Sorio, Fabio Puglisi

**Affiliations:** 1Department of Medicine (DAME), University of Udine, 33100 Udine, Italy; giacomo.pelizzari.med@gmail.com (G.P.); lorenzo.gerratana@cro.it (L.G.); lucia.bortot@cro.it (L.B.); fabio.puglisi@cro.it (F.P.); 2Unit of Medical Oncology and Cancer Prevention, Department of Medical Oncology, Centro di Riferimento Oncologico di Aviano (CRO), IRCCS, 33081 Aviano, Italy; dlombardi@cro.it (D.L.); sscalone@cro.it (S.S.); rsorio@cro.it (R.S.); 3Division of Molecular Oncology, Centro di Riferimento Oncologico di Aviano (CRO), IRCCS, 33081 Aviano, Italy; mnicoloso@cro.it (M.N.); gbaldassarre@cro.it (G.B.); 4Unit of Gynecological Oncology, Centro di Riferimento Oncologico di Aviano (CRO), IRCCS, 33081 Aviano, Italy; ggiorda@cro.it

**Keywords:** platinum-sensitive ovarian cancer, bevacizumab, PARP-inhibitors

## Abstract

Introduction: Targeted agents such as bevacizumab (BEV) or poly (ADP-ribose) polymerase inhibitors (PARPi) which have been added as concomitant or maintenance therapies have been shown to improve progression-free survival (PFS) in patients with platinum-sensitive recurrent ovarian cancer (PS rOC). In the absence of direct comparison, we performed a network meta-analysis considering *BRCA* genes status. Methods: We searched PubMed, EMBASE, and MEDLINE for trials involving patients with PS rOC treated with BEV or PARPi. Different comparisons were performed for patients included in the PARPi trials, according to *BRCA* genes status as follows: all comers (AC) population, *BRCA* 1/2 mutated (BRCAm), and *BRCA* wild type patients (BRCAwt). Results: In the overall population, PARPi prolonged PFS with respect to BEV (hazard ratio (HR) = 0.70, 95% CI 0.54–0.91). In the *BRCA* mutated carriers, the PFS improvement in favor of PARPi appeared to be higher (HR = 0.46, 95% CI 0.36–0.59) while in BRCAwt patients the superiority of PARPi over BEV failed to reach a statistically significance level (HR = 0.87, 95% CI 0.63–1.20); however, according to the SUCRA analysis, PARPi had the highest probability of being ranked as the most effective therapy (90% and 60%, for PARPi and BEV, respectively). Conclusions: PARPi performed better as compared with BEV in terms of PFS for the treatment of PS rOC, especially in BRCAm patients who had not previously received PARPi.

## 1. Introduction

Epithelial ovarian cancer (EOC) with its most frequent high-grade serous histology is the leading cause of death among gynecological malignancies in developed countries [1].

Despite surgical and medical efforts in the upfront treatment, about 70% of EOC patients have a disease relapse within five years after diagnosis. Recurrent EOC is still an incurable condition with a median overall survival ranging from 12 to 24 months [2]. Three out of four patients experience disease recurrence at least six months after the last platinum dose, a time frame which is called the platinum-free interval that has been considered to be a clinical surrogate of ovarian cancer sensitivity to platinum salts [3]. Until very recently, in patients with a clinically defined platinum-sensitive recurrent EOC, the gold standard therapy was mainly based on rechallenge with a platinum-based regimen. Recently, treatment of platinum-sensitive recurrent EOC has been improved by the addition of the anti-VEGF antibody bevacizumab or the poly (ADP-ribose) polymerase inhibitors (PARPi) to the platinum-based regimen. As a matter of fact, results from three phase III trials have shown an improvement in progression-free survival (PFS) with the addition of bevacizumab in association with a platinum-based chemotherapy, and then as maintenance therapy, as compared with chemotherapy alone, a benefit observed also in patients previously exposed to first-line bevacizumab-containing therapy [4,5,6]. On the basis of this premise, in 2016, the FDA approved bevacizumab for the treatment of PS rOC in association with gemcitabine or paclitaxel as a platinum companion. Recently, the therapeutic armamentarium for platinum-sensitive recurrent EOC has taken a step forward by approving three drugs belonging to the PARPi class (i.e., olaparib, niraparib, and rucaparib). These drugs have been tested in phase II and phase III placebo-controlled trials as maintenance therapy after partial or complete response to a platinum-based treatment, showing a benefit to progression-free survival in the overall population of recurrent EOC patients, especially in those with a germline (gBRCA) or somatic (sBRCA) mutation in the *BRCA* 1/2 genes [7,8,9,10,11,12]. In many cases, recurrent EOC is a chemo-sensitive disease which is manageable with several lines of new and older anticancer therapies and as a consequence, treatment strategy is now a challenging field for the gynecologic oncologist. Some of these new agents, such as niraparib and veliparib, have shown remarkable antitumoral activity also in heavily pretreated patients and, at a lower dose, they could be integrated with radiotherapy or chemotherapy [13,14]. In clinical oncology, patients with advanced solid tumors are generally treated with the most active drug that has demonstrated the greatest clinical benefit in delaying disease progression. In this perspective, defining the best treatment after the first platinum-sensitive recurrence, is still an unmet need in the absence of trials that directly compare the two available maintenance strategies. Moreover, if the presence of *BRCA* mutation is considered to be a predictive factor for PARPi benefit, currently, for the vast majority of patients with a *BRCA* wild type (BRCAwt) status, there are no predictive biomarkers for PARPi or for bevacizumab that could guide the clinicians’ choice between the two target therapies [15]. In this scenario, we performed a network meta-analysis (NMA) to evaluate the differences in terms of efficacy between bevacizumab and PARPi therapies for women with platinum-sensitive recurrent EOC, according to *BRCA* genes status.

## 2. Results

After the selection process, eight randomized trials were included in the NMA for a total of 3402 patients. The role of bevacizumab was investigated by three trials, (n = 3, 1563 patients) among which the trial by Pignata et al., although still not published in extenso, was the only trail testing bevacizumab beyond progression, i.e., in patients previously exposed to bevacizumab in the first-line setting [6]. The other five studies concerned maintenance therapy with PARPi (n = 5, 1839 patients), specifically olaparib, rucaparib, and olaparib. There was only one trial by Oza et al. that tested a PARPi (olaparib) in concomitance to chemotherapy, and then as maintenance therapy [10]. The selected studies are summarized in Table 1. 

Overall, three treatment arms were identified representing the distinct maintenance strategies, bevacizumab, PARPi, and surveillance post chemotherapy (CT). The networks between trials in the AC population and in the other subgroups are presented in Figure 1. In the AC population, PARPi improved progression-free survival as compared with bevacizumab (HR = 0.70, 95% confidence interval (CI) 0.54–0.91). In the *BRCA* 1/2 mutated (BRCAm) patients, the gain in progression-free survival reached by a PARPi therapy was greater (HR = 0.46, 95% CI 0.36–0.59). In the subgroup of BRCAwt patients, the superiority of PARPi over bevacizumab failed to reach a statistically significance level (HR = 0.87, 95% CI 0.63–1.20) but despite this, PARPi had the highest probability of being classified as the most effective therapy considering the SUCRA values (90% and 60%, for PARPi and bevacizumab, respectively) (Table 2). Forest plots are reported in Figure 2. 

## 3. Discussion

In the present study, we performed an indirect comparison between the available bevacizumab and PARPi-based trials in a platinum-sensitive recurrent EOC patient setting through a network meta-analysis to test which strategy could obtain the highest gain in terms of progression-free survival. According to this indirect estimation, PARPi performed the best in terms of a progression-free survival benefit for the treatment of platinum-sensitive recurrent EOC, regardless of *BRCA* genes status. An added value of our work is that patients who received PARPi were further analyzed in the following three different groups, according to the available data on *BRCA* gene status: AC, BRCAm, and BRCAwt patients. As expected, given the efficacy of these drugs in ovarian cancer with homologous recombination (HR) deficiency, the best performance of PARPi was observed in the BRCAm subgroup. For BRCAwt patients, the benefit of PARPi over bevacizumab was not statistically significant, but PARPi maintenance therapy had the highest likelihood of being ranked as the best treatment in terms of efficacy, according to SUCRA values. 

Almost invariably, relapsed ovarian cancer is not curable with patients undergoing several lines of platinum and non-platinum therapy for advanced disease. In the last decade, the armamentarium of medical treatment options for patients experiencing a platinum-sensitive recurrence has been enriched with two targeted agents, the antiangiogenic drugs and PARPi. The addition of the anti-VEGF antibody bevacizumab during platinum chemotherapy, and then as maintenance therapy, has been demonstrated to increase progression-free survival as compared with platinum alone, even in patients who had previously received bevacizumab as a part of the first-line treatment, after primary surgery. In the same setting, maintenance therapy with a PARPi after a complete/partial response to platinum therapy provided a progression-free survival gain as compared with the placebo, in particular for patients with a deficit in the HR pathway, as for *BRCA* 1/2 mutation carriers. In an effort to optimize the treatment strategy, which agent between bevacizumab and PARPi should be added to a platinum chemotherapy to maximize the clinical benefit for these patients is still an unsolved clinical question, given the absence of direct comparisons from randomized trials. The present work supports the use of PARPi maintenance therapy over bevacizumab in PS rOC. If the benefit of a PARPi over bevacizumab could appear predictable in patients with *BRCA* 1/2 mutation, the same was not foregone for the BRCAwt subgroup. Therefore, although through indirect evidence, the NMA indicates that, in the case of recurrent disease, patients with BRCAwt status should preferentially undergo PARPi maintenance therapy. Our results are in line with those presented by Feng et al. who performed an indirect comparison among three different maintenance strategies available for patients with ovarian cancer, i.e., PARPi, chemotherapeutic agents and angiogenesis inhibitors (among which there were pazopanib, cediranib, and nintedanib). In that work, maintenance therapy with PARPi showed better performance as compared with angiogenesis inhibitors in terms of PFS (HR 0.73, 95% credibility interval 0.63–0.86) [16]. A peculiar aspect of our study was the focus on bevacizumab which is the most common angiogenesis inhibitor and the only one approved in the maintenance setting of relapsed ovarian cancer. In addition, we performed the analysis according to *BRCA* gene status. Our work and that of Feng et al. also share the limit of a comparison between trials with different designs. In fact, PARPi-based trials enrolled patients in response to platinum-therapy, whereas patients who progressed during chemotherapy were eligible in the bevacizumab-based trials. Due to the heterogeneity among trial populations, meta-analysis is not suitable for comparing the different maintenance strategies in patients with platinum-sensitive ovarian cancer. For this purpose, a tailored comparison by head-to-head trial is, therefore, required, and the results from indirect comparisons of trials should only be considered hypothesis generators.

Nevertheless, bevacizumab continues to play a role in the management of platinum-sensitive recurrent EOC and should be added to platinum therapy in patients who present a high burden of disease at relapse where a prompt tumor shrinkage could reduce disease-related symptoms. In fact, as emerged in the OCEANS trial, the addition of bevacizumab increased objective response rate of about 20% as compared with a platinum combination alone, improving chemotherapy performance [4]. 

Which platinum doublet should be combined with bevacizumab is influenced by the licensed combinations. In April 2017, the European Medicine Agency (EMA) extended the indication of bevacizumab in combination with carboplatin/gemcitabine or carboplatin/paclitaxel for patients with first recurrence of platinum-sensitive epithelial ovarian cancer [17]. Despite this, recent evidence from the ENGOT-ov18/AGO-OVAR 2.21 trial has shown a better performance in progression-free survival for the combination of carboplatin plus pegylated liposomal-doxorubicin and bevacizumab over carboplatin plus gemcitabine and bevacizumab (median progression-free survival months 13.3 vs. 11.7 months, HR 0.80, 95% CI 0.68–0.96, *p* = 0.0128) [18]. The possible impact of chemotherapy backbone in the performance of the bevacizumab-based trial has not been evaluated in the present work and trials that compare different chemotherapy regimens with bevacizumab have not been included because of the lack of a bevacizumab-free control arm. 

Other considerations to be taking into account in the choice of the better maintenance strategy, are the different toxicities profiles of these targeted agents. Bevacizumab has a manageable toxicity, and it has a specific side effects profile due to its mechanism of action. The most common adverse events include hypertension, proteinuria, hemorrhages, thrombotic events, poor wound healing, and gastrointestinal perforation. As a consequence, patients who have previously experienced or are at a higher risk of developing these side effects should not receive bevacizumab and could opt for a PARPi if indicated [19]. Maintenance therapy with PARPi is generally well tolerated. The most common Grade ≥3 toxicities attributed to the class effects of these drugs include anemia and fatigue, but there are distinct safety and toxicity profiles among the different PARPi [20] that could be, in part, explained by the different trapping potency on PARP1 enzyme [21]. Among the others, talazoparib has demonstrated the most remarkable trapping potency and it is registered for use in advanced breast cancer at a lower dose than niraparib, olaparib, and rucaparib [22]. 

Currently, the treatment dilemma of recurrent ovarian cancer is further influenced by the changing landscape of the first-line therapy. The SOLO-1 trial established a new standard of care in *BRCA* 1/2 mutation carriers in which olaparib was demonstrated to reduce the risk of progression by about 70% as compared with a placebo [23]. In the up-front setting, niraparib has also been shown to improve progression-free survival over a placebo, in a population at high-risk of recurrence; the benefit was reached in *BRCA* 1/2 mutated patients and in *BRCA* wild type patients with a positive HR deficiency score assessed by the Miriad myChoice HRD test [24,25]. Similar results have been reported by the combination therapy with olaparib and bevacizumab in the PAOLA1 GINECO/ENgOT-ov25 trial [26] and in the VELIA trial where platinum therapy was combined with veliparib [27]. As more patients have access to a PARPi first-line therapy, the treatment strategy will become important and trials testing the best therapeutic sequence are needed. Moreover, future studies should explore the mechanism of resistance to PARPi; first clues have been reported for acquired mutations on the *BRCA* gene that restored the reading frame, thereby conferring resistance to PARPi [28]. Finally, since PARPi seem to play a synergistic activity with various biologic agents, results from trials testing novel strategies (i.e., combination of immune check point inhibitors, small molecules, and antiangiogenic agents with chemotherapy) are eagerly awaited [29].

Notwithstanding the interesting insights emerging from the present study, some limitations mainly deriving from the different designs of the trials included in the network analysis should be acknowledged (Figure 3).

Firstly, in the bevacizumab-based trials the randomization was performed at evidence of recurrence before starting chemotherapy plus/minus bevacizumab. Conversely, in trials testing the PARPi-based maintenance therapy, the randomization occurred after at least four cycles of a platinum-based chemotherapy and only in patients experiencing a partial/complete response to treatment. Considering that progression-free survival is calculated from randomization to evidence of disease progression or death, the progression-free survival in bevacizumab-based trials also includes the time when chemotherapy was administered (about four months), potentially increasing the performance of the bevacizumab maintenance therapy. However, in the trials testing PARPi, the inclusion of patients in complete or partial response to platinum could have selected a population with a better prognosis in which response to PARPi could be more probable. Moreover, in the trials testing PARPi, patients who experienced a progression within six months from the beginning of chemotherapy were not included, and these represented about 20% of randomized patients according to the OCEANS trial [4]. Another consideration is that, in the bevacizumab-based trials, the progression-free survival assessment in the subgroup of BRCAm and BRCAwt patients was not available. As a consequence, in our analysis according to *BRCA* status, data from the bevacizumab-based trials were considered as for the all comers. In terms of outcomes, our analysis was focused on efficacy evaluation using progression-free survival. Analysis on overall survival was not performed due to the lack of mature and published data for all trials included. Finally, this work did not analyze the differential toxicity profile of the two maintenance strategies that could be crucial in the treatment choice.

## 4. Materials and Methods

### 4.1. Search Strategy and Study Selection

A comprehensive research and analysis of studies was conducted using PubMed, EMBASE, and MEDLINE. The research strategy included key words as the following: “platinum sensitive” AND “ovarian cancer” AND “bevacizumab” OR “poly (ADP-ribose) polymerase inhibitors” AND “randomized controlled trial”. Meeting proceedings of the American Society of Clinical Oncology (ASCO) and the European Society of Medical Oncology (ESMO) were searched to find phase II and phase III trials not yet published in extenso. Randomized phase II or phase III clinical trials comparing platinum-based chemotherapy plus bevacizumab or PARPi (i.e., olaparib, niraparib, rucaparib, talazoparib, and veliparib) for the treatment of platinum-sensitive recurrent EOC published or presented from 1 January 2009 to 1 January 2019 were considered. Trials testing the activity or the efficacy of other anti-angiogenic drugs than bevacizumab (e.g., cediranib and trebananib) were not included. Single-arm trials were excluded. Studies in which progression-free survival was not the primary or secondary endpoint were excluded, as well as trials conducted in a first-line setting. Each trial was screened by two independent reviewers (R.S. and S.S.). A third reviewer (M.B) was consulted for controversies and for definitive approval. The risk of biases of selected trials was assessed using Review Manager (RevMan) version 5.3 software (The Nordic Cochrane Centre, The Cochrane Collaboration, 2014, Copenhagen, Denmark) (Appendix A). The trials selection process was summarized in the PRISMA plot (Figure 4).

### 4.2. Data Collection and Statistical Analysis

A frequentist NMA was carried out utilizing the graph-theoretical methodology by Rucker et al. [30]. A random effects model was implemented. The primary outcome was to compare the efficacy of bevacizumab vs. PARPi therapy in woman with platinum-sensitive recurrent EOC, in terms of progression-free survival (defined as the time between randomization and disease recurrence or death). Data on safety, response rate, and survival were not analyzed. Data were collected only from studies reporting hazard ratios (HRs) and 95% confidence intervals (CI) for progression-free survival. Other data extracted from the analyzed studies were sample size, germline *BRCA* mutational status, year of publication, and first author. The primary analysis was performed in the all comers (AC) patient population. A further analysis was conducted to assess the efficacy of bevacizumab vs. PARPi according to *BRCA* mutational status. To rank the size effect of treatments, surface under the cumulative ranking curve (SUCRA) value was applied [31]. All analyses were performed using R Statistical Software (version 3.5.1) along with the “netmeta” package (version 0.9-8). Differences in PFS among the PARPi trials that tested olaparib, rucaparib, and niraparib were not explored as this was beyond the scope of the present study. The study was a network meta-analysis of anonymous aggregate data without any direct or indirect intervention on patients thus, it was not required any ethical approval. 

## 5. Conclusions

These indirect comparisons of prospective trials have provided the first evidences demonstrating the superiority of PARPi maintenance therapy over bevacizumab maintenance therapy for platinum-sensitive recurrent EOC. According to our results, PARPi therapy should be the preferred choice for platinum-sensitive ovarian cancer patients.

## Figures and Tables

**Figure 1 ijms-21-03805-f001:**
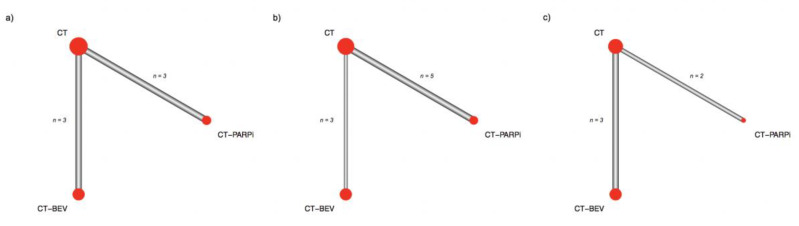
Network geometry. Edges thickness is proportional to the number of direct treatment comparisons. Node size is proportional to the number of patients considered for a given treatment. (**a**) All comer population; (**b**) *BRCA* mutated patients; (**c**) *BRCA* wild type patients.

**Figure 2 ijms-21-03805-f002:**
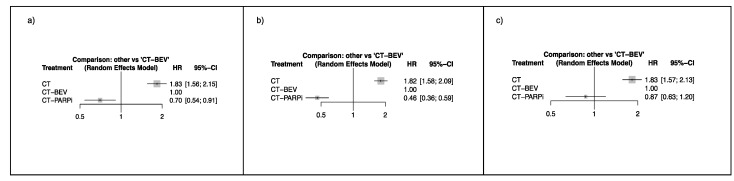
Hazard ratios (HR) of progression-free survival (PFS) for PARPi-based trials (CT-PARPi) as compared with bevacizumab-based trials (CT-BEV) and chemotherapy (CT) alone without maintenance. (**a**) All comers population; (**b**) *BRCA* mutated; (**c**) *BRCA* wild type patients.

**Figure 3 ijms-21-03805-f003:**
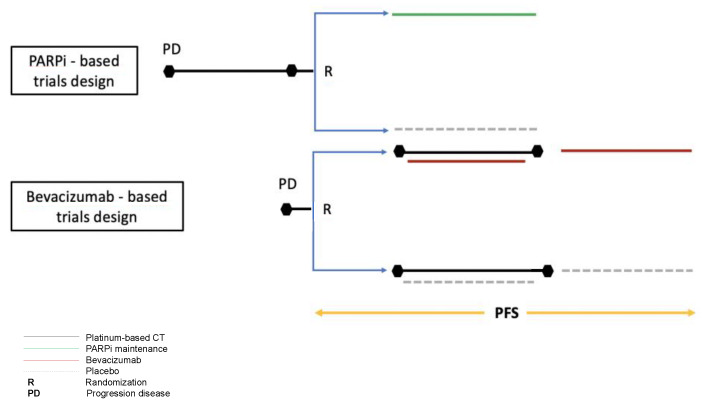
Different designs between PARPi and bevacizumab pivotal trials. In the bevacizumab-based trial randomization was performed at disease progression, before chemotherapy started; In the PARPi-based trials randomization occurred in the case of partial or complete response to platinum therapy.

**Figure 4 ijms-21-03805-f004:**
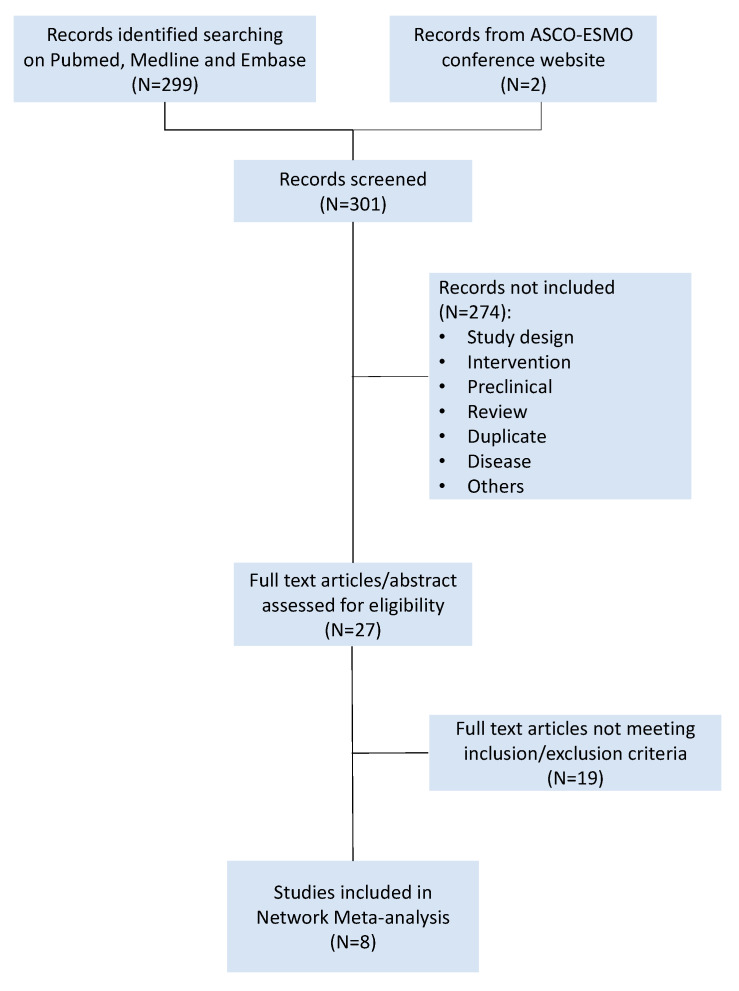
PRISMA flowchart of platinum-sensitive recurrent ovarian cancer randomized controlled trials.

**Table 1 ijms-21-03805-t001:** Studies included in the network meta-analysis. Data on all comers (AC), *BRCA* mutated (BRCAm), and *BRCA* wild type (BRCAwt) subgroups are reported, arranged in different rows.

Authors	Design	Population	Primary Endpoint	N. Patients Randomized	Treatment Arms	HR for PFS	CI (95%)	*p* Value
**Coleman R. et al. (2017) GOG 0213**	Phase III	Recurrent EOC, PFI ≥ 6 m, prior anti-VEGF allowed	OS PFS (secondary)	A: 337 B: 337	A: Carboplatin AUC5 + Paclitaxel 175 mg/m^2^ q21 ×6 cycles B: Carboplatin AUC5 + Paclitaxel 175 mg/m^2^ + bevacizumab 15 mg/kg q21 × 6 cycles followed by bevacizumab 15 mg/kg q21 maintenance until PD	0.628 for B	0.534–0.739	0.0001
Aghajanian C et al. (2012) OCEANS	Phase III	Recurrent EOC, PFI ≥ 6 m, prior anti-VEGF not allowed	PFS	A: 242 B: 242	A: Carboplatin AUC4 d1 + Gemcitabine 1000 mg/m^2^ d1-8 q21 × 6 cycles B: Carboplatin AUC4 d1 + Gemcitabine 1000 mg/m^2^ d1-8 + bevacizumab 15 mg/kg d1 q21 × 6 cycles followed by bevacizumab 15 mg/kg q21 maintenance until PD	0.484 for B	0.388–0.605	0.0001
Pignata S. et al. ASCO 2018 MITO-16	Phase III	Recurrent EOC, PFI ≥ 6 m, treated with anti-VEGF in 1°line	PFS	A: 203 B: 202	A: Carboplatin + Paclitaxel/Gemcitabine/PLD q21 × 6 cycles. B: Carboplatin + Paclitaxel/Gemcitabine/PLD + bevacizumab 15 mg/kg q21 x 6 cycles followed by bevacizumab 15 mg/kg q21 maintenance until PD	0.51 for B	0.41–0.64	0.001
Ledermann J. et al. (2012) STUDY-19	Phase II	Recurrent HGSOC, PFI ≥ 6 m, treated with a median of 2 platinum-based regimens All comers	PFS	A: 129 B: 136	A: Placebo B: Olaparib 400 mg twice daily until PD	0.35 for B	0.25–0.49	0.001
Ledermann J. et al. (2014) STUDY-19	Phase II	Recurrent HGSOC, PFI ≥ 6 m, treated with a median of 2 platinum-based regimens BRCAm	PFS	A: 62 B: 74	A: Placebo B: Olaparib 400 mg twice daily until PD	0.18 for B	0.10–0.31	0.0001
Ledermann J. et al. (2014) STUDY-19	Phase II	Recurrent HGSOC, PFI ≥ 6 m, treated with a median of 2 platinum-based regimens BRCAwt.	PFS	A: 61 B: 57	A: Placebo B: Olaparib 400 mg twice daily until PD	0.54 for B	0.34–0.85	0.0075
Oza M. et al. (2015)	Phase II	Recurrent EOC, PFI ≥ 6 m. All comers	PFS	A: 81 B: 81	A: Carboplatin AUC5 + Paclitaxel 175 mg/m^2^ q21 × 6 cycles B: Carboplatin AUC5 + Paclitaxel 175 mg/m^2^ q21 + olaparib 200 mg d1-10 q21 × 6 cycles » Olaparib 400 mg twice daily until PD	0.51 for B	0,34–0,77	0.0012
Oza M. et al. (2015)	Phase II	Recurrent EOC, PFI ≥ 6m BRCAm	PFS	A: 21 B: 20	A: Carboplatin AUC5 + Paclitaxel 175mg/m^2^ q21 × 6 cycles B: Carboplatin AUC5 + Paclitaxel 175mg/m^2^ q21 + olaparib 200 mg d1-10 q21 × 6 cycles » Olaparib 400 mg twice daily until PD	0.21 for B	0.08–0.55	0.0015
Mirza M.R. et al. (2016) **NOVA**	Phase III	Recurrent HGSOC, PFI ≥ 6m, at least 2 platinum-based regimens	PFS	A: 65 B: 138	gBRCAm A: Placebo B: Niraparib 300 mg twice daily until PD	0.27 for B	0.173–0.410	0.0001
**Mirza M.R. et al.** (2016) **NOVA**	Phase III	Recurrent HGSOC, PFI ≥ 6m, at least 2 platinum-based regimens	PFS	A: 116 B: 234	not-gBRCAm A: Placebo B: Niraparib 300 mg twice daily until PD	0.45 for B	0.338–0.607	0.0001
Coleman R et al. (2017) **ARIEL-3**	Phase III	Recurrent HGS or endometrioid OC PFI ≥ 6m. at least 2 platinum-based regimens All comers	PFS	A: 189 B: 375	A: Placebo B: Rucaparib 600 mg	0.36 (ITT population)	0.30–0.45	0.0001
Coleman R et al. (2017) **ARIEL-3**	Phase III	Recurrent HGS or endometrioid OC, PFI ≥ 6m, at least 2 platinum-based regimens BRCAm	PFS	A: 66 B: 130	A: Placebo B: Rucaparib 600 mg	0.23	0.16–0.34	0.0001
Pujade-Lauraine E et al. (2017) **SOLO-2**	Phase III	Recurrent HGS or endometrioid OC with g/sBRCA 1/2 m, PFI ≥ 6 m, at least 2 platinum-based regimens	PFS	A: 99 B: 196	A: Placebo B: Olaparib 300 mg in two 150 mg tablets, twice daily, until PD	0.30 for B	0.22–0.41	0.0001

EOC: epithelial ovarian cancer; HGSOC: high grade serous ovarian cancer; PFI: platinum-free interval; OS: overall survival; PFS: progression -free survival; AUC: area under the time-concentration curve; PD: progression disease; HR: hazard ratio; CI: confidence interval.

**Table 2 ijms-21-03805-t002:** SUCRA values by different treatments in BRCAwt patients.

Treatment Efficacy		
Treatment	SUCRA	Rank
**PARPi**	90%	1
**BEV**	60%	2
**CT**	0%	3

SUCRA: surface under the cumulative ranking value; PARPi: poly (ADP-ribose) polymerase inhibitors; BEV bevacizumab; CT: chemotherapy without maintenance treatment.

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
