# Peer review of "Bevacizumab or PARP-Inhibitors Maintenance Therapy for Platinum-Sensitive Recurrent Ovarian Cancer: A Network Meta-Analysis"

_ijms, 2020, doi:10.3390/ijms21113805_

Round 1

Reviewer 1 Report

In this article, the authors performed a network meta-analysis (NMA) to evaluate differences in terms of efficacy between bevacizumab and PARPi therapy in women with platinum-sensitive recurrent EOC, according to BRCA genes status.

Network meta-analysis is used to compare three or more treatments for the same condition [Rücker 2015]. However, in this article, as authors mention in the Discussion, different groups are compared. Bevacizumab-treated patients include those who did not respond to therapy, although PARPi-treated patients consist of those who respond to therapy alone. Also, PFS cannot be compared as the time at randomization was different. Feng et al. (2019) have reported “Comparison of PARPis with Angiogenesis Inhibitors and Chemotherapy for Maintenance in Ovarian Cancer: A Network Meta-Analysis (Adv Ther).” The authors need to comment on this paper and discuss how the heterogeneity of data should be managed.

In Figure 2, “nodes” are absent and the difference in “width” cannot be recognized. The authors should improve this figure.

I think the effects of PARPis, i.e., olaparib, niraparib, and rucaparib, could be compared with the use of this technique (NMA).

Reviewer 2 Report

In this review Bartoletti M et al, performed a network meta-analysis to evaluate differences in terms of efficacy between bevacizumab and PARP inhibitors therapy in women with platinum-sensitive recurrent epithelial ovarian cancer, according to BRCA genes status. The paper is straightforward and concise. Definitely deserves to be published and is a valuable contribution to the “International Journal of Molecular Sciences”. Some minor flaws need to be addressed before publication.

Minor points:

[1] “Introduction”, Lines 51-57:

“Recently, the therapeutic armamentarium for platinum-sensitive recurrent EOC has made another step forward with the approval of three molecules belonging to PARPi class (i.e. olaparib, niraparib and rucaparib). These drugs have been tested in phase II-III placebo-controlled trials as maintenance therapy after partial or complete response to a platinum-based treatment, showing a benefit in progression-free survival in the overall population of recurrent EOC patients, especially in those with a germline (gBRCA) or somatic (sBRCA) mutation in the BRCA 1/2 genes [7],[8],[9],[10],[11]”.

In terms of rucaparib, chronologically was approved by the FDA in December 2016 and by the EMA in May 2018 for the treatment of high-grade serous EOC, fallopian tube, or primary peritoneal carcinoma patients with germline or somatic BRCA mutations, relapsed after at least two chemotherapy lines. Notably, the efficacy was demonstrated by a pooled analysis of two multicenter, single-arm clinical trials; study 10 and ARIEL 2 study.

Recommended reference: Oza AM, et al. Antitumor activity and safety of the PARP inhibitor rucaparib in patients with high-grade ovarian carcinoma and a germline or somatic BRCA1 or BRCA2 mutation: Integrated analysis of data from Study 10 and ARIEL2. Gynecol Oncol. 2017 Nov;147(2):267-275.

[2] “Introduction”, Lines 57-59:

“In many cases, recurrent EOC is a chemo-sensitive disease manageable with several lines of new and older anticancer therapies and as a consequence, treatment strategy is now a challenging field for the gynecologic oncologist.”

Please, make here a comment about the role of veliparib in this setting. There are available phase I and II studies of veliparib monotherapy in heavily pretreated patients with relapsed epithelial ovarian cancer, which demonstrate a considerable efficacy with an acceptable toxicity profile. The reported objective response rates range between 26% and 65%. Furthermore, veliparib has been extensively investigated in combination with various cytotoxic agents, based on its favorable toxicity profile. There is also preclinical evidence-based rationale for combination treatment of low-dose fractionated whole abdominal radiation with veliparib.

Relevant reference: Boussios S, et al. Veliparib in ovarian cancer: a new synthetically lethal therapeutic approach. Invest New Drugs. 2020 Feb;38(1):181-193.

[3] “Results”, Lines 110-111:

“Only one trial by Oza et al. tested a PARPi (olaparib) in concomitance to chemotherapy and then as maintenance[10].”.

In addition to that, the phase III VELIA trial presented at the 2019 ESMO congress. The study demonstrated that patients with high-grade serous epithelial ovarian cancer, fallopian tube, or primary peritoneal carcinoma experienced a 32% reduction in the risk of progression or death with frontline combination veliparib plus carboplatin and paclitaxel followed by veliparib maintenance treatment.

Recommended reference:

Coleman RL, et al. Veliparib with First-Line Chemotherapy and as Maintenance Therapy in Ovarian Cancer. N Engl J Med. 2019 Dec 19;381(25):2403-2415.

[4] “Discussion”, Lines 142-144:

“As expected, given the efficacy of these drugs in ovarian cancer with Homologous Recombination (HR) deficiency, the best performance of PARPi was observed in the BRCAm subgroup.”

At that point, should be also highlighted the phenomenon of PARP inhibitors resistance. The restoration of homology-directed DNA repair through secondary reversion mutations is the most common identified mechanism of resistance. The restoration of BRCA activity starts from BRCA-deficient and chemo-sensitive cells as a result of several mutations that are induced by platinum agents. This initial restored clone expands in the setting of treatment-specific selective pressure. Furthermore, loss of 53BP1 function by either mutation or downregulation accelerates the BRCA1-independent end-resection and provides PARP inhibitor resistance. Epigenetic silencing or accelerated protein synthesis and degradation could also lead to decreased expression of PARP enzymes, followed by PARP inhibitors resistance.

Relevant reference: Boussios S, et al. PARP Inhibitors in Ovarian Cancer: The Route to "Ithaca". Diagnostics (Basel). 2019 May 18;9(2). pii: E55.

[5] “Discussion”, Lines 188-190:

 “Maintenance therapy with PARPi is generally well tolerated. The most common grade ≥ 3 toxicities attributed to the class effects of these drugs include anemia and fatigue, but there are distinct safety and toxicity profiles among the different PARPi[18].”.

Please, explain further the rationale of the distinct toxicity profiles among PARP inhibitors. It has been shown that increased PARP trapping is associated with high myelosuppression, which results in variation of the recommended doses across PARP inhibitors. The most potent PARP trapping agent talazoparib has been investigated at daily dose of 1 mg, as compared to 300 mg or greater for the remaining PARP inhibitors.

[6] “Discussion”, Lines 194-197:

“In the up-front setting, also niraparib has shown to improve progression-free survival over placebo in a population at high-risk of recurrence; the benefit was reached in BRCA 1/2 mutated patients and in BRCA wild-type patients with a positive HR deficiency score assessed by Miriad myChoice HRD test[20],[21].”.

Apart from PRIMA study (ref 20) in the up-front setting, it should be mentioned here the Quadra trial for late-line treatment. Based on the results of the Quadra trial, the FDA approved in October 2019 niraparib for patients with advanced homologous recombination deficient epithelial ovarian cancer, or primary peritoneal cancer treated with at least three prior chemotherapy regimens.

Recommended reference: Moore KN, et al. Niraparib monotherapy for late-line treatment of ovarian cancer (QUADRA): a multicentre, open-label, single-arm, phase 2 trial. Lancet Oncol. 2019 May;20(5):636-648.

[7] “Discussion”, Lines 199-200:

“As more patients access to a PARPi first line therapy, treatment strategy will become important and trials testing the best therapeutic sequence are needed.”.

Kindly, make a statement here about the combination strategies. Several biologic agents have been studied in combination with PARP inhibitors, including anti-angiogenics, immune checkpoint inhibitors, phosphoinositide 3-kinase (PI3K), protein kinase B (AKT), mammalian target of rapamycin (mTOR), WEE1, mitogen-activated protein kinase (MEK), and cyclin dependent kinase (CDK) 4/6 inhibitors, as well as the standard chemotherapy in the epithelial ovarian cancer. This therapeutic approach may sensitize epithelial ovarian cancers with de novo or acquired HR proficiency to PARP inhibitors.

Recommended reference: Boussios S. Combined Strategies with Poly (ADP-Ribose) Polymerase (PARP) Inhibitors for the Treatment of Ovarian Cancer: A Literature Review. Diagnostics (Basel). 2019 Aug 1;9(3). pii: E87.

Round 2

Reviewer 1 Report

The revised version has been improved.  

Line 219. 95% confidence interval

Author Response

Addressed